# School Gardens: Initial Training of Future Primary School Teachers and Analysis of Proposals

**José Orenes Cárceles [1,2,\*], Gabriel Enrique Ayuso Fernández [1], Manuel Fernández-Díaz [1] and José María Egea Fernández [3]**

1   Department of Science Education, Faculty of Education, Universidad de Murcia, 30100 Murcia, Spain; ayuso@um.es (G.E.A.F.); manuel.fernandez2@um.es (M.F.-D.)
2   CEIP—Public School Ntra. Sra. de las Mercedes, 30836 Murcia, Spain
3   Faculty of Biology, Universidad de Murcia, 30100 Murcia, Spain; jmegea@um.es
\*   Correspondence: jose.orenes@um.es; Tel.: +34-690-38-78-01

**Abstract:** This paper describes a training proposal for future teachers in the design, management, and use of school gardens as an educational resource. During the 2020/2021 academic year, future teachers in the 4th grade of Primary Education (last year of the University Degree) received theoretical-practical classes to develop teaching-learning activities in the area of Sciences that they implemented in a Primary school. This training proposal and the research activities designed and implemented in the school garden by future teachers were analysed using three criteria: curricular contents covered, competency richness and structure and content. From the training programme implemented, we highlight an increase in the motivation of students towards learning to use this resource; because it is learned "in situ" in the school garden, it is possible to carry out outdoor work, sharing natural resources through cooperative work and improving relationships. Regarding the design of activities proposed by the future teachers, there is a predominance of the use of observation and classification processes and a deficit of other scientific competences, which implies the need for greater specific initial training on school gardens.

**Keywords:** school garden; scientific processes; educational proposals; initial teacher training

## 1. Introduction

The school garden (SG, hereafter) is a resource with great educational potential, being useful for integrating different ways of learning [1] both topics included in the curriculum and those that are not [2–4]. For example, this resource is very useful for the approach of contents related to healthy habits and biodiversity [5], mathematics [6], environmental protection [7,8], geology [9,10], or the appreciation of natural richness and beauty [11], among others. The school garden (SG) has become part of the landscape of educational institutions, similar to the science laboratory, library, or computer room [12]. The value of the garden as a didactic resource depends on the skill with which it is managed and used in the teaching-learning process, to understand cause and effect relationships, to practise and apply what is learned, to use it as a laboratory in different subjects, to take advantage of the resources of the environment, and simultaneously to prepare children for life [13]. The evaluations made by the students of Primary School Degree, as well as those of practising teachers who had the experience of conducting these activities in school, show a high level of satisfaction with the use of the SG resource [14].

Various investigations have shown that student teachers as well as active teachers have pointed to the need for training in SG [14–16]. In this context, it is necessary to train future teachers in their design, maintenance, and didactic use so as not to rely on the teachers' ability alone, making their existence, maintenance, and practice a common way of working in our educational institutions.

We believe that any training plan must deal with both theoretical and practical contents offering examples of useful proposals for teachers that serve as examples with which they can design new material appropriate to different educational contexts and the contents of the Primary Education curriculum, in all its areas. To a large extent, most of the activities designed and implemented in the SGs contribute to the development of cross-cutting contents, mainly in Education for Health and Sustainability [17–22].

Training in the design and maintenance of SG is not easy given the wide range of subjects involved (plants, photosynthesis, nutrition, soil-compost, fertilisers, irrigation systems, seeds, crops, pests, agricultural techniques, biocultural memory, agroecology...), and is a task that begins from the student's first contact with the garden and never ends, as in the SG we learn something new and useful every day. As observed in [23], all the knowledge that is acquired can be used in the future.

Likewise, training in the use of SG as an educational resource is also a complex task; each area has a specific didactic that describes how to design appropriate activities for the work of its contents. In accordance with these perspectives, and although an interdisciplinary vision of the SG has been promoted, in the contents related to the area of Sciences for Primary Education, an approach based on exploratory activities has been proposed.

Consequently, with these considerations, this work has the following objectives:

Objective 1. Elaborate, implement, and evaluate a basic theoretical-practical preparation proposal to train the undergraduate students to design educational proposals for the work of the SG in Primary Education.

Objective 2. Analyse the proposals of the undergraduate students and verify the effectiveness of the same for an implementation of a teaching program for the use of the SG in Primary.

## 2. Materials and Methods

### 2.1. Description of the Sample

Related to objective 1, a training plan was implemented for 37 students, aged between 21 and 41 years, of the Faculty of Education of the University of Murcia (Spain), in the 4th year of Primary School Degree during the 2020/2021 academic year in the specialty "Educational resources for school and free time" on the design, management, and use of SG as an educational resource. This training plan was implemented in three phases.

In the first phase, students received a two-hour session in their classroom. Table 1 summarises the contents discussed, the development followed, and the sources of support for the proposals used. Subsequently, another session was held, also of two hours, in the orchards of the Faculty of Biology of the University of Murcia (Figure 1) that consisted of the installation of drip irrigation, fertilising soil, and autumn planting: beans, lettuce, broccoli, onion, and cauliflower (Table 2).

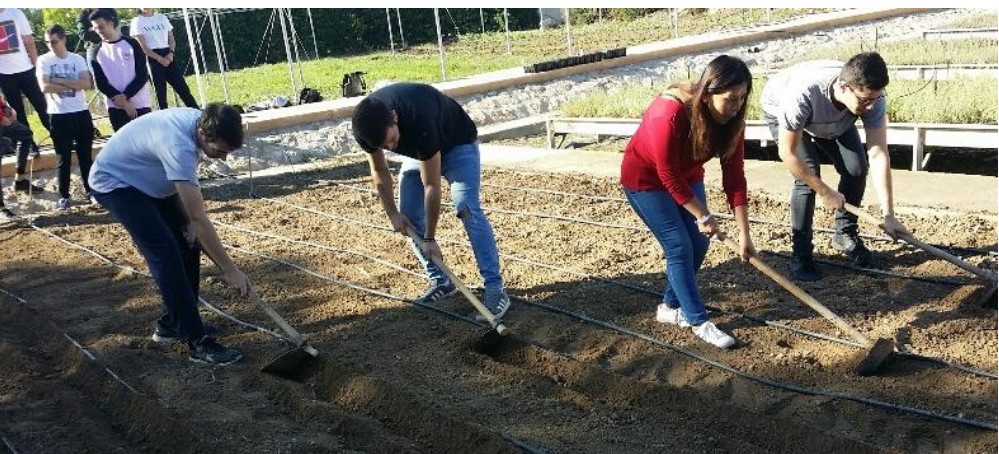

**Figure 1.** Students preparing the orchard.

**Table 1.** Characteristics of the training plan implemented in phase one in the classroom.

| Contents | Development | Place Duration Resources | Sources of Support |
|---|---|---|---|
| SG features. Disadvantages | Presentation of previous experiences and knowledge. | Class 25 min Questionnaire Pooling | |
| Contents that can be worked on cross-cutting and curricular areas. | Presentation of examples of activities: Education in values, for Health, Environmental, Interdisciplinary . . . | Class 10 min PowerPoint presentation | |
| Area of Sciences; scientific processes with exploratory activities | Analysis of an activity example. The students agree on the format in the activity design. | Class 10 min "My plant is growing" | Activity designed by 2nd Grade students |
| The SG. Concept, types of SG | Presentation of examples: SG on the ground, in pots, on tables, vertical . . . | Class 10 min PowerPoint | [24] |
| Management: organisation of the SG, implementation of activities and necessary resources. | 1st Meeting with the educational community. Formation of the "Motor Group". 2nd Planning and elaboration of the objectives and the work plan. 3rd Evaluation of resources 4th Distribute tasks. Activity Log | Class 10 min Debate PowerPoint presentation | [25] |
| Maintenance: people in charge, schedules, and communication channels. | Participating students. Teachers involved and their time distribution. Activities. Workshops. Maintenance during school holidays. Bulletin board. | Lesson 5 min Brainstorming PowerPoint | [25] |
| Most frequent tasks to perform in a SG | Cleaning, materials, crops, agricultural techniques, vegetable collection, compost, language workshop, poems and stories, biodiversity, aromatic workshop, sculptures with fruits and vegetables, scarecrows, cooking workshop and healthy consumption habits, seed bank, market, Carnival, Biodiversity Festival, open day, recovery of knowledge and flavours: biocultural memory. | Class 10 min Brainstorming PowerPoint presentation | [25] |
| Crops. Crop rotation. | A table was set out with the most cultivated species in the SG and the characteristics related to their planting. | Lesson 5 min PowerPoint presentation | [25] |
| Improvement of health. Pest control. | In the classroom. Presentation with slides of the main pests and their control. | Class 10 min PowerPoint presentation | [25] |
| Biocultural memory. | In the SG, children are introduced to an agro-ecological education: principles. | Class 25 min PowerPoint presentation | [26] |

In a second phase, after the training period, the Primary Grade student teachers worked in 9 different groups (4–5 students per group) collaboratively carrying out the planning and design of an exploratory activity for a chosen level of Primary Education using

the SG resource, consistent with the training received. Later, this activity was implemented with Primary Education students in a school in Murcia (Spain).

**Table 2.** Characteristics of the training plan implemented in phase one in the orchard.

| Theoretical-Practical Contents Covered | Development | Place Duration Resources | Sources of Support |
|---|---|---|---|
| Materials and tools used in the SG | Description and use of the main materials and tools. | Orchard 10 min Handling | |
| Soil and composting. | Orchard fertiliser. The importance of composting and how to put it into practice. | Orchard 20 min Spreading organic fertiliser | [27] |
| Seeds. | The students were informed of different ways of obtaining seeds and their conservation. Distribution of seeds and plants. | Orchard 15 min Collecting seeds: peas, beans, lettuce, onion . . . | [25] |
| Irrigated land. | Installation of drip irrigation system. | Orchard 20 min | |
| Autumn planting. Obtaining plant materials. Irrigation | Planting of beans (seeds), lettuce, broccoli, onion, and cauliflower (plants): spacing, depth, number of seeds . . . Figure 1 | Orchard 40 min | [25] |

In a third phase, the activities proposed by the groups of undergraduate students used with Primary students are discussed in the large group. The assessments made on the implementation by the 9 regular teachers of the Primary groups in which the interventions have been carried out are analysed. Each group of undergraduate students presents their proposal. Two of the researchers of this paper lead the session in which the positive aspects and the aspects to be improved of each proposal are analysed.

### 2.2. Characteristics of the Training Plan

The theoretical-practical contents have been developed in the training plan in phase one in the classroom (Table 1) and in the orchard (Table 2).

### 2.3. Instruments for the Analysis of Students' Work

Related to Objective 2, an analysis of the activities designed and implemented was carried out by two of the authors of this paper using several different criteria.

First, the curricular contents worked on in the different activities designed by the undergraduate students in the area of Sciences and other areas (Table 3).

**Table 3.** Instrument for the registration of information on the contents covered in the activities proposed by the groups of undergraduate students.

| Activity | Contents Covered | |
|---|---|---|
| | **Main Concepts:** | |
| | Scientific competences | |
| **Activity name:** | Procedures planned: | ☐ Observation ☐ Classification ☐ Predictions/hypotheses ☐ Experimental design / ☐ Measuring/data analysis ☐ Establish conclusions and ☐ Communicate results |
| | Expected attitudes to work: | |
| Group, components: | Other areas involved in the activity: | |

Secondly, an analysis of the competence richness of the activities proposed by the groups of undergraduate students before being implemented in the classroom (Table 4) [28], whereby the two reviewing authors indicate whether each of the aspects to be analysed is carried out.

**Table 4.** Analysis of the competence richness of the activities proposed by the groups of undergraduate students.

| Aspects to Be Analysed in the Proposals |
| :---: |
| **Context** |
| 1. Does the proposal or learning situation refer to a real or everyday context, is it carried out in the SG, and/or is it socially relevant for the students? |
| 2. Is it an open proposal? (Does it allow differentiated contributions, contrast of ideas, issue hypotheses, offer various experimental designs . . . ?) |
| 3. Does it favour the emergence of ideas that the students have in relation to the facts or phenomena that we can observe in the SG, to work from them and make them evolve? |
| **Science** |
| 4. Is the proposal or learning situation significant, does it refer to previous ideas of the students, is it functional, and does it allow the use of scientific processes? |
| 5. Does the activity encourage students to pose research questions that help them to understand and interpret the facts and phenomena observable in the SG? |
| 6. Does it help to interpret facts or phenomena of the SG from its complexity, to obtain learning using scientific models? |
| **Ways of working** |
| 7. Does the experimental work involve the use of different instruments, tools (including ICT), agricultural techniques, or materials, both those of daily use in the SG and more specific ones relating to school science? |
| 8. Do you work from "good questions" guiding the teaching process where students are the protagonists contributing ideas rather than "teacher explanations"? |
| 9. Do you work taking into account both the knowledge of the students, the teachers, the dialogue we establish with the environment in the SG . . . as well as other sources of information (books, newspapers or magazines, websites . . . )? |
| 10. Does it require both individual work and responsibility and cooperative work in pairs or in groups that leads to talking, listening, arguing, convincing, agreeing . . . ? |
| 11. Does it stimulate the work of scientific values such as: making predictions, looking for evidence, systematic doubt, perseverance, scientific rigour . . . as well as scientific interest and imagination? |
| **Self-regulation and autonomy** |
| 12. Does it help students to reflect on what they do, reason, and communicate by using different language forms (oral, written, graphic, physical, arts . . . ) using scientific reports, oral presentations, murals, videos, SG blog, school website . . . ? |
| 13. Do you encourage the autonomy, initiative, and self-regulation of students so that they are fully involved and are aware of their learning? |
| **Transfer and action** |
| 14. Do you have to apply knowledge already acquired from other areas by relating it to other situations, and make new learning, applying knowledge in different contexts: sustainability, environment, health, coexistence, use of ICT, biocultural memory, agroecology . . . ? |
| 15. Does the activity encourage students to intervene in their local environment by putting into practice knowledge, values, and norms of coexistence? |

Source: Adapted from CESIRE-CDEC (2009) [28].

On the other hand, Table 5 represents an analysis of the structure and content of the proposed activities. The two evaluators will reflect whether the following dimensions are observed in the students' work.

**Table 5.** Structure and content of the activities designed.

| Aspects of the Proposals to Be Analysed |
|:---:|
| 1. Find a title for the activity |
| 2. Define the contents to be worked on according to the sequence of contents established in the section (Conceptual, procedural, and attitudinal contents) |
| 3. Select didactic objectives (educational achievements) of the curriculum related to the SG |
| 4. Arouse interest and motivation for the resolution of the problem raised in the activity |
| 5. Describe the activity with a logical sequence of actions on the part of the students and the teacher |
| 6. Ask for previous ideas or hypotheses about the problem posed |
| 7. Select materials and instruments necessary to do the activity |
| 8. Think and describe the experimental procedure that must be followed to obtain evidence and solve the problem posed in the activity |
| 9. Plan how to collect and organise the data |
| 10. Plan how to analyse data and establish results |
| 11. Provide for how to draw conclusions from the investigation |
| 12. Review the elaborated text |
| 13. Establish different groupings, plan the time allocated to the task according to the possibilities of the students and the necessary aids |
| 14. Design the appropriate instruments for evaluating the learning and teaching process |

Finally, the 9 regular teachers of the Primary groups in which the research is carried out evaluated the implementations of the proposals of the undergraduate students. For this, they used the criteria established in Table 6 on the attitudes and personal qualities of undergraduate students and the teaching action implemented.

**Table 6.** Aspects analysed (by regular Primary teachers) on the implementation of the proposed activities carried out by undergraduate students.

| Attitudes and Personal Qualities |
|:---:|
| 1. Precision and good use of the language. |
| 2. Voice projection. |
| 3. Class management and behavior control. |
| 4. Creation of an affective and working climate in the classroom. |
| 5. Empathy with students. |
| 6. Ability to adapt to unforeseen situations. |
| 7. Professional attitude. Responsibility. |
| 8. Professional attitude. Initiative. |
| 9. Capacity for self-assessment. |
| 10. Positive response to advice. |
| **Teaching action** |
| 1. Clear specification of purposes and objectives. |
| 2. Adaptation of the activities to the intended objectives, in particular, to the intended knowledge. |

**Table 6.** *Cont.*

| Attitudes and Personal Qualities |
|:---:|
| 3. Knowledge of the subjects involved. |
| 4. Preparation and use of resources or teaching materials. |
| 5. Clarity in instructions and explanations. |
| 6. Type, variety and balance in the activities. |
| 7. Rhythm and use of time. |
| 8. Attention to differences. |
| 9. Active participation of students. |
| 10. Achievement of the objectives set |

## 3. Results and Discussion

### 3.1. Results Related to the Training Programme for Undergraduate Students (Objective 1)

From our training programme developed on the design, management, and use of SG in Primary Education, we can point to the following results:

For all the students, it served as a basis for gaining knowledge and using the SG as a resource in Primary Education. After the training phase, each group designed and implemented an activity using the SG as an educational resource.

The methodology used was participatory, allowing the students to interact with each other and with the teacher; it was based on their ideas and their training needs, which meant a constant motivation throughout the training process.

Some training tasks were carried out in the Faculty of Biology Garden, which further increased the interest of the undergraduate students to know how to use this educational resource; the future teachers learned from it "in situ" [29].

Cooperative and small group work improved the training programme; each group chose to perform a different task, and each student learned their mission responsibly [24].

In the implementation phase of the designed activities, all the groups of Primary Education students and their tutors demanded the realisation of these activities for their group.

Most of the activities were carried out in an environment outside the classroom, perceiving and manipulating the phenomenon or object of study outdoors.

The work of the scientific processes, using exploratory activities, did not achieve the purpose established among the undergraduate students because in the results it is observed that they only designed activities where the scientific processes of observation and classification predominate, evidencing that they need more didactic training in designing activities under this profile.

The time used, given the large amount of content addressed in the training programme, was limited (2 + 2 h). The students expressed the need for greater training in many of the contents addressed, coinciding with [14,26,30].

### 3.2. Results Related to the Educational Proposals Developed by Undergraduate Students to Be Used in Primary Education Classrooms (Objective 2)

The productions designed by the undergraduate students and subsequently implemented in a Primary school were analysed according to three different criteria:

1. Regarding the curricular contents for the area of the Sciences and other areas of the 9 activities designed by the undergraduate students in Primary Education. The results are collected separately in three phases ("cycles") of Primary Education in Spain: 6–8, 8–10, and 10–12 years of age (Tables 7–9).

**Table 7.** Analysis of the contents treated in the activities designed by the undergraduate students of the Primary Education, 6–8 years old.

| Activity | Contents Worked for the Primary Education, Schoolchildren between 6 and 8 Years of Age | | |
|---|---|---|---|
| | **Main Concepts: Vegetables, Seeds, Flowers. Tools. The Sense Organs** | | |
| Activity name: **1. "We learn with the senses"** | Procedures planned: | Scientific competences | |
| | | × Observation × Classification ☐ Predictions/hypotheses ☐ Experimental design | ☐ Measuring/data analysis ☐ Establish conclusions and ☐ Communicate results |
| Group, components: 7–8 years old | Expected attitudes to work: Teamwork. Respect comrades. Caring for the environment | | |
| | Other areas involved in the activity: | | |
| Activity name: **2. "We grow our plant"** | Main concepts: Plants, their structure and physiology | | |
| | Procedures planned: | Scientific competences | |
| | | × Observation × Classification ☐ Predictions/hypotheses ☐ Experimental design | ☐ Measuring/data analysis ☐ Establish conclusions and ☐ Communicate results |
| Group, components: 7–8 years old | Expected attitudes to work: Development of work habits and responsibility. Habits of respect and care | | |
| | Other areas involved in the activity: Artistic Education with the drawing of plants from the garden | | |
| Activity name: **3."Planting aromatics"** | Main concepts: Germination. Seed. Aromatic plants. Tools | | |
| | Procedures planned: | Scientific competences | |
| | | × Observation × Classification ☐ Predictions/hypotheses × Experimental design | ☐ Measuring/data analysis ☐ Establish conclusions and ☐ Communicate results |
| Group, components: 7–8 years old | Expected attitudes to work: Respect for plants, environment, and care. Importance of aromatics in SG. | | |
| | Other areas involved in the activity: | | |

**Table 8.** Analysis of the contents covered in the activities designed by the undergraduate students for the Primary Education, 8–10 years old.

| Activity | Contents Worked on for the Primary Education, Schoolchildren between 8 and 10 Years of Age | | |
|---|---|---|---|
| Activity name: **4. "Insect houses"** | Main concepts: Living beings, characteristics, types (earthworms, ladybugs . . . ) Biodiversity in the SG. Pollination. | | |
| | Procedures planned: | Scientific competences | |
| | | × Observation × Classification ☐ Predictions/hypotheses ☐ Experimental design | ☐ Measuring/data analysis ☐ Establish conclusions and × Report results |
| Group, components: 8–9 years old | Expected attitudes to work: Respect and care of living beings. Importance of biodiversity in the SG | | |
| | Other areas involved in the activity: | | |

**Table 8.** *Cont.*

| Activity | Contents Worked on for the Primary Education, Schoolchildren between 8 and 10 Years of Age | | |
|---|---|---|---|
| **Activity name: 5. "Self-control. Radish planting"** | Main concepts: Materials and safety standards. Planting radishes. | | |
| | Procedures planned: | Scientific competences | |
| | | × Observation<br>× Classification<br>× Predictions/hypotheses<br>× Experimental design | × Measure/data analysis<br>× Draw conclusions and<br>× Report results |
| **Group, components: 8–9 years old** | Expected attitudes to work: Value cooperative work. Care for your own safety and colleagues. Taking care of tools and proper use. Habits of respect and care. | | |
| | Other areas involved in the activity: | | |
| **Activity name: 6. "Themed scarecrow"** | Main concepts: Recycling. Unwanted birds in the SG. The story. | | |
| | Procedures planned: | Scientific competences | |
| | | × Observation<br>☐ Classification<br>☐ Predictions/hypotheses<br>☐ Experimental design | ☐ Measuring/data analysis<br>☐ Establish conclusions and<br>☐ Communicate results |
| **Group, components: 8–9 years old** | Expected attitudes to work: Raise awareness about the importance and benefits generated by recycling in the preservation of the environment, reduction, and reuse. | | |
| | Other areas involved in the activity: Artistic Education with the creation of a scarecrow. Spanish Language and Literature when making stories. | | |
| **Activity name: 7. "Game: Who am I?"** | Main concepts: Living beings of the SG. Functions of nutrition, relationship, and reproduction. | | |
| | Procedures planned: | Scientific competences | |
| | | × Observation<br>× Classification<br>☐ Predictions/hypotheses<br>☐ Experimental design | ☐ Measuring/data analysis<br>☐ Establish conclusions and<br>☐ Report results |
| **Group, components: 8–9 years old** | Expected attitudes to work: Awareness towards the preservation and care of living beings. Raise awareness of the importance of biodiversity in the SG. | | |
| | Other areas involved in the activity: | | |
| **Activity name: 8. "A mini vegetable greenhouse"** | Main concepts: Living beings, diversity, and functions that characterise it. Germination and development of plants. Seeds. | | |
| | Procedures planned: | Scientific competences | |
| | | × Observation<br>× Classification<br>× Predictions/hypotheses<br>× Experimental design | × Measure/data analysis<br>× Draw conclusions and<br>× Report results |
| **Group, components: 9–10 years old** | Expected attitudes to work: Development of work habits. Effort and responsibility. Teamwork. I respect my peers. Care for the environment. Respect and care for living beings. | | |
| | Other areas involved in the activity: | | |

We can see in Tables 7–9 that the future teachers covered curriculum contents in the area of the Sciences in all the activities designed, as requested, but also included in two of them, in an interdisciplinary way, contents of Spanish Language and Artistic Education.

At the same time, they also worked on cross-cutting issues such as: Care for the Environment, Moral and Civic Education, Education for Peace and Coexistence, and Health very much in accordance with [17].

**Table 9.** Analysis of the contents covered in the activities designed by the undergraduate students for the Primary Education, 10–12 years old.

| Activity | Contents Worked for the Primary Education, Schoolchildren between 10 and 12 Years of Age | | |
|---|---|---|---|
| | **Main Concepts: Own Vocabulary. Plants. Plant Types: Angiosperms and Gymnosperms. Parts of a Plant. The News.** | | |
| Activity name: ***9. "Our friends the plants"*** | Procedures planned: | Scientific competences | |
| | | × Observation<br>× Classification<br>☐ Predictions/hypotheses<br>☐ Experimental design | ☐ Measuring/data analysis<br>☐ Establish conclusions and<br>× Report results |
| Group, components: 10–11 years old | Expected attitudes to work: Respect and care for plants. Respect and value my own and others' creations. | | |
| | Other areas involved in the activity: Artistic Education with the drawing of plants from the garden. Spanish Language and Literature when creating a news story about my plant. | | |

We observe that although the activities, in principle, are required as exploratory activities incorporating the usual scientific processes with the students who carry them out, only in two activities was this requirement observed. In the other seven, the work of only two processes is observed: observe and classify the phenomena or objects studied. As indicated in [31], the garden is consolidated as a resource used to promote observation as the main scientific procedure.

2.  Results of the analysis of the activities designed by the groups of undergraduate students regarding their competency richness adapted to the SG resource (Table 10).

**Table 10.** Analysis of the competence richness of the activities proposed by the groups of undergraduate students.

| Aspects to Be Analysed in the Proposals | Yes | No |
|---|---|---|
| **Context** | | |
| 1. Does the proposal or learning situation refer to a real or everyday context, is it carried out in the SG, and is it socially relevant for the students? | 9 | 0 |
| 2. Is it an open proposal? (Does it allow differentiated contributions, contrast of ideas, issue hypotheses, make different experimental designs . . . ?) | 7 | 2 |
| 3. Does it favour the emergence of ideas that the students have in relation to the facts or phenomena that we can observe in the SG, to work from them and make them evolve? | 8 | 1 |
| **Science** | | |
| 4. Is the proposal or learning situation significant, refer to the previous ideas of the students, is it functional, and allows the work of scientific processes? | 5 | 4 |
| 5. Does the activity encourage students to pose research questions that help them understand and interpret the facts and phenomena observable in the SG? | 5 | 4 |
| 6. Does it help to interpret facts or phenomena of the SG from its complexity, to obtain learning using scientific models? | 5 | 4 |
| **Ways of working** | | |
| 7. Does the experimental work involve the use of different instruments, tools (including ICT), agricultural techniques or materials, both those of daily use in the SG and more specific ones of school science? | 4 | 5 |
| 8. Do you work from "good questions" guiding the teaching process where students are the protagonists contributing ideas rather than "teacher explanations"? | 4 | 5 |

**Table 10.** *Cont.*

| Aspects to Be Analysed in the Proposals | Yes | No |
|---|---|---|
| **Context** | | |
| 9. Do you work taking into account both the knowledge of the students, the teachers, the dialogue we establish with the environment in the SG . . . as well as other sources of information (books, newspapers or magazines, websites . . . )? | 7 | 2 |
| 10. Does it require both individual work and responsibility and cooperative work in pairs or in groups that leads to talking, listening, arguing, convincing, agreeing . . . ? | 7 | 2 |
| 11. Does it stimulate the work of scientific values such as: making predictions, looking for evidence, systematic doubt, perseverance, scientific rigour . . . as well as scientific interest and imagination? | 4 | 5 |
| **Self-regulation and autonomy** | | |
| 12. Does it help students to reflect on what they do, reason, and communicate using different language forms (oral, written, graphic, physical, arts . . . ) using scientific reports, oral presentations, murals, videos, SG blog, school website . . . ? | 6 | 3 |
| 13. Do you encourage the autonomy, initiative, and self-regulation of students so that they involve and are aware of their learning? | 6 | 3 |
| **Transfer and action** | | |
| 14. Do you have to apply knowledge already acquired from other areas by relating it to other situations, and make new learning, applying knowledge in different contexts: sustainability, environment, health, coexistence, use of ICT, biocultural memory, agroecology . . . ? | 8 | 1 |
| 15. Does the activity encourage students to intervene in their local environment by putting into practice knowledge, values, and norms of coexistence? | 8 | 1 |
| **Total** | 93 | 42 |

Source: Adapted from CESIRE-CDEC (2009) [28].

Table 10 shows a great wealth of competence (93 'yes' responses) in all the questions raised. Among the most interesting data in this table, it should be noted that all the proposals referred to a real and daily context in the SG (question 1), [32], thus increasing their motivation [14,33].

On the other hand, most of the activities designed to encourage students to present their ideas on the work topics observed in the SG (question 3) are based on the previous knowledge of the students on which to build new learning (question 14) and promote intervention in the environment by putting into practice knowledge, values, and norms of coexistence (question 15) [34–37].

The least observed competency criteria refer to the low involvement of experimental work using different instruments, tools (including ICT), agricultural techniques, or materials whether they are used daily in the SG or of more specific use in schools (question 7); not to work from "good questions" that guide the teaching process with the students being the protagonists of their learning and the teacher who contributes ideas more than "explanations" (question 8), and that the work of scientific values such as: making predictions, looking for evidence, systematic doubt, perseverance, scientific rigour . . . as well as scientific interest and imagination (question 11) coinciding with the results of Tables 7–9.

3.  Results of the analysis of the activities designed by the undergraduate students in Primary Education regarding their structure and contents requested as learning activities for the area of Sciences (Table 11).

Regarding the data table in Table 11 on the content and structure of the activities, all have an adequate and suggestive title (9 out of 9 designed). In all of them, the contents covered are well defined and the proposed objectives are selected. The degree of motivation for the activities is also high in all of them.

Its design based on the previous ideas of the students or hypotheses posed about the problem is very low, only observed in 1 of the 9 activities.

**Table 11.** Analysis of the structure and content of learning activities implemented by the undergraduate students of 4th Grade of Primary Education.

| Aspects of the Proposals to Be Analysed | YES | NO |
|---|---|---|
| 1. Find a title for the activity | 9 | 0 |
| 2. Define the contents to be worked on according to the sequence of contents established in the section (conceptual, procedural, and attitudes) | 9 | 0 |
| 3. Select didactic objectives (educational achievements) of the curriculum related to the SG | 9 | 0 |
| 4. Arouse interest and motivation for the resolution of the problem raised in the activity | 9 | 0 |
| 5. Describe the activity with a logical sequence of actions on the part of the students and the teacher | 8 | 1 |
| 6. Ask for previous ideas or hypotheses about the problem posed | 1 | 8 |
| 7. Select materials and instruments necessary to do the activity | 9 | 0 |
| 8. Think and describe the experimental procedure that must be followed to obtain evidence and solve the problem posed in the activity | 1 | 8 |
| 9. Plan how to collect and organise the data | 2 | 7 |
| 10. Plan how to analyse data and establish results | 1 | 8 |
| 11. Provide for how to draw conclusions from the investigation | 2 | 7 |
| 12. Review the elaborated text | 1 | 8 |
| 13. Establish different groupings, plan the time allocated to the task according to the possibilities of the students and the necessary aids | 9 | 0 |
| 14. Design appropriate learning and teaching process assessment tools | 3 | 6 |
| **Total** | 73 | 53 |

Regarding the material and instruments to be used in the activities, they are well defined in all of them. Note that most of this material was provided by the undergraduate students (acquiring or building it) because it did not exist in the Primary school. This is a common inconvenience in many schools.

On the description of the experimental procedure to be followed to solve the problem posed, since the problem does not arise in most of the activities, it is only observed in 1 of the 9 activities.

Most do not plan sufficiently how to collect and organise data (2 out of 9) or how to analyse and establish results (only 1 out of 9); nor the way to draw conclusions from the investigation (only 2 out of 9).

In all activities, different groupings were established.

Finally, mechanisms for the evaluation of learning and the development of teaching were only established in 3 out of 9 activities.

4.  Results of the evaluation of the implementation of the proposals of the undergraduate students by the usual Primary teachers.

The nine Primary teachers of the groups of students of this educational level valued very positively the implementations carried out by the undergraduate students. Especially, they highlighted the interest and motivation of future teachers both in aspects related to attitudes and personal qualities, as well as their teaching action.

This information, obtained from the completion of the surveys by the regular teachers of the Primary groups, was analysed in the final session with all the undergraduate students and two of the researchers who were the authors of this study. In this final session, undergraduate students shared the characteristics of their respective proposals. The aspects to be improved and that have been referred to in the previous sections were also analysed.

## 4. Conclusions and Educational Implications

We believe it is relevant, in a training plan, that students participate in the development of their activities, can put into practice knowledge, values, and coexistence rules and learn under the constructivist paradigm "in situ", in the SG.

Learning in the garden through participation supposes a high intrinsic motivation in the students [16,32], since it is framed in real and daily situations that interest them, as indicated by the studies of [31] where the children express fun, satisfaction, or well-being for the realisation of the activities and the involvement of the students is encouraged [13]; as happened with the student teachers.

The work in the SG is an opportunity for the implementation in the classroom of active and experiential methodologies in which students can learn science doing science. Working on research activities, in the area of Sciences, with this resource is very appropriate because the SG is a living laboratory. Therefore, it is necessary to have a specific material, although this supposes, sometimes, an added difficulty that can lead to the abandonment of designs of learning activities for other types of activities.

The SG resource makes it possible to carry out outdoor work, sharing natural resources through cooperative work and improving relations between boys and girls as indicated [33–36].

Any training plan with the SG should attend to collaborative work respecting values and rules of coexistence as it is a characteristic of this resource.

The SG is a useful resource for both interdisciplinary work and for the development of cross-cutting themes: Environmental Education, Health Education, Education for Coexistence, Literature (for example, through short poem contests), Geology (brief information about the bedrock and/or sediments in which plants grow), Geometry, and Arithmetic (measure SG, count seeds) [6,9,10]

Finally, future teachers need examples of activities where scientific values are developed such as: making predictions, looking for evidence, systematic doubt, perseverance . . . as well as scientific interest and imagination. This denotes a lack of training in the design of this type of activity where scientific processes related to initiation to scientific activity in Primary Education. Offering more examples of learning activities and implementing them in future training programmes will give them the knowledge they need to be able to use them in their future interventions.

To optimise the use of school gardens as an educational tool, it is necessary to promote practical courses on school gardens and food sovereignty aimed at the "training of trainers", as well as technical assistance for the management of the garden and the development of activities [15,25].

**Author Contributions:** Conceptualisation, J.O.C., G.E.A.F. and J.M.E.F.; methodology, J.O.C. and G.E.A.F.; software, M.F.-D.; validation, J.O.C., G.E.A.F., M.F.-D. and J.M.E.F.; formal analysis, G.E.A.F.; research, J.O.C., G.E.A.F., M.F.-D. and J.M.E.F.; resources, J.O.C., G.E.A.F. and M.F.-D.; data curation, G.E.A.F.; writing—preparation of the original draft, J.O.C. and G.E.A.F.; writing—proofreading and editing, J.O.C. and G.E.A.F.; visualisation, J.O.C. and G.E.A.F.; supervision, J.O.C. and G.E.A.F.; project management, J.O.C. and G.E.A.F. All authors have read and agreed to the published version of the manuscript.

**Funding:** This research received no external funding.

**Informed Consent Statement:** Informed consent was obtained from all subjects involved in the study.

**Data Availability Statement:** The data presented in this study are available on request from the corresponding author.

**Conflicts of Interest:** The authors declare no conflict of interest.

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
