# Peer review of "School Gardens: Initial Training of Future Primary School Teachers and Analysis of Proposals"

_education, doi:10.3390/educsci12050303_

Round 1

Reviewer 1 Report

I would like to thank the authors for an undoubtedly interesting research problem. This study does not look at similar experiences in other schools, perhaps even in other countries. Please provide similar examples and analyze the literature. I cannot be sure of the novelty of this approach. Also, the study sample is too small in my opinion. There are no restrictions, no information about how this research can be developed.

Reviewer 2 Report

Dear Authors,

congratulations on the paper! I have read it with great interest, because I myself use the geological garden in my teaching. Thank you for many good ideas!

The subject matter you raise in your article is rarely mentioned in the literature, as evidenced by the lack of English-language works in the Introduction. I have indicated a few examples of literature. In the table, I quoted errors and a suggestion for improvement. 

Good luck!

Reviewer 3 Report

The paper presents a methodology to develop and assess learning activities in the school garden for primary school teachers.It is a training proposal and  presents the results achieved by a class of "future teachers" in developing these activities and determining the scientific content included and explored in them. 

As a research paper, it has some flaws, since its goals and objectives are not exactly research goals. It presents and describes (well) a methodology, which can be applied elsewhere, and is therefore replicable, but one can't really infer whether or not its application is useful for pupils. It would be interesting to be able to assess its results in practice.

The training plan is interesting and presents a useful wide focus. 

Sources used are all Spanish. It would definitely be important to scope other works outside o the Spanish context in the development of training proposals.

Round 2

Reviewer 1 Report

The article has been finalized and may be accepted for publication